# Reciprocal Relationship between Parents' School- and Home-Based Involvement and Children's Reading Achievement during the First Year of Elementary School

Sabrina Bonanati [1,*] and Charlott Rubach [2]

1   Deparment of Educational Psychology, Paderborn University, 33098 Paderborn, Germany
2   School of Education, University of California, Irvine, CA 92697, USA; crubach@uci.edu
*   Correspondence: sabrina.bonanati@upb.de

**Abstract:** Reading is an essential competence students learn in school. One question is how parents can support their children and their reading competence, particularly at the beginning of elementary school. Guided by this question, this study investigated the longitudinally reciprocal relationship between parental school- and home-based involvement with children's reading competence. We also tested whether school- and home-based involvement mediated the relationship between structural context variables (e.g., migration background) and reading competence. A total of 254 parent–child dyads answered a questionnaire at two measurement points, i.e., at the beginning and the end of the first grade in elementary school. Home-based involvement and reading competence were negatively, reciprocally related to each other. Furthermore, we found a negative association between reading competence at the beginning of grade 1 and the relative change in school-based involvement at the end of grade 1. No mediation effects of school- and home-based involvement in the relation between structural context variables and reading competence were found. This paper provides a deeper understanding of the complex interrelations of the family–school partnership during the first school year.

**Keywords:** parental involvement; school-based involvement; home-based involvement; reading achievement; first grade

## 1. Introduction

The shared responsibility among schools and families is a critical factor for students' academic development in the first years of their school career. Shared responsibility implicates that "teachers and parents [ . . . ] share common goals for their children that are achieved most effectively when teachers and parents work together" [1] (p. 121). Everyday collaboration can be manifested in parental school- and home-based involvement. Both have been revealed to be significant determinants of children's school-related outcomes, such as motivation and achievement in school [2–5]. School-based involvement is described as parents' engagement in family–school partnerships [6]. Parents who participate in school, for example, by attending parent–teacher conferences, or volunteering in school activities, positively influence their children's development in school [7]. Home-based parental involvement includes parental behavior and learning-related interactions with the child outside school (e.g., help with homework, supervision, reading together, monitoring learning activities, [7,8]).

Although it is assumed that parental involvement influences children's academic development, we know little about the effect of both parental school- and home-based involvement on children's reading performance at the beginning of elementary school [9]. The transition from kindergarten to elementary school implicates a start for parent and teacher partnerships when parents and teachers do not know each other and need to establish communication and cooperation structures. Knowledge about the effectiveness of

collaboration for a child's positive academic development might help teachers adjust their offers and communication to parents and vice versa.

Furthermore, previous research that focused on the impact of parental involvement on children's achievement often used measures of overall academic success. Investigating domain-specific competencies, such as reading competencies, can lead to more detailed information for schools and teachers to understand the relevance of their partnership in different subjects and might strengthen subject-specific family–school partnerships [10]. Next to the impact of school- and home-based involvement on children's achievement, prior studies also demonstrated that children's achievement is an antecedent for parents to become more involved in schools [11]. For example, parents of children with lower reading achievement levels become more engaged in their child's school life than parents of children with higher levels of reading achievement [4,12,13].

Adding on these findings, the following study aimed to investigate the reciprocal relationship between children's achievement and parental school- and home-based involvement during the first year in elementary school. We focused on the domain of reading and addressed the need to clarify the direction of the investigated relationships [14]. As parental involvement and children's achievement often are determined by structural context variables (e.g., parents' migration and educational background), we will further investigate the relationship between these context variables with children's reading achievement mediated by parents' school- and home-based involvement [15,16].

### 1.1. School- and Home-Based Involvement and Its Influence on Children's Achievement

Based on Hoover–Dempsey and Sandler's [8,17] model of parental involvement process and Eccles and Harold's [11] model of influences on and consequences of parent involvement in schools, it can be assumed that parental involvement influences children's academic beliefs (e.g., motivation), achievement, and competencies. However, the question of how parents become involved in their children's school life and impact their children's academic achievement need to be considered.

Parental involvement is a multidimensional construct [14]. The most common differentiation, which only concerns parental behaviors, is between school- and home-based involvement [3,7,18]. The two main forms of school-based involvement are volunteering and communicating. A good communication between teachers and parents can be established when parents are helping at the school or attending school events [3,7,19,20]. Scholars assume that regular communication leads to a more effective home-based involvement (e.g., help with homework, monitoring child's learning process, [3,5,18]). Another explanatory factor of parental school- and home-based involvement are child characteristics [8,11,17]. One child characteristic is the parents' perception of their child's school performance [21]. When parents observe their child having trouble with homework, they might adjust their homework help by increasing it. This assumption also implies that parents who believe their child is performing well at school decrease their school- or home-based involvement because no help is needed.

### 1.2. The Cross-Sectional Perspective on Positive and Negative Relationships between School- and Home-Based Involvement and Children's Achievement

To summarize, children's achievement can be assumed as both an outcome and also as an explanatory factor for parental involvement [22]. In the following literature review, we will first report findings from cross-sectional studies and meta-analyses. Then, we will focus on longitudinal studies, which might give insight into the reciprocal relationship between parental involvement and children's achievement. If available, we will report findings on school- and home-based involvement separately.

At first, meta-analyses of cross-sectional studies have shown positive effects of parents' school-based involvement on students' achievement across subjects [2,3,23,24]. Findings on the impact of parental home-based involvement were inconsistent. Hill and Tyson [3] did not find a statistically significant association between general home-based involvement

and general children's achievement. Jeynes [23] reported a significant positive effect of general home-based involvement for children's reading achievement. Furthermore, studies have postulated that the effect sizes of the impact of parental involvement (school- and home-based) were larger when student achievement was operationalized as across or within-subject Grade Point Average (GPA). Studies using test scores of specific subjects, such as reading or mathematics, found the effect sizes of the effect of parents' involvement to be smaller [3,23].

There is also evidence that children's achievement influences their parents' involvement. However, children's achievement has primarily been shown to be negatively associated with parental school- and home-based involvement [4,12,13,25]. One interpretation might be that parents who perceived that their children had difficulties with the school work increased their school- and home-based involvement.

### 1.3. The Longitudinal Perspective on the Relationship between School- and Home-Based Involvement and Children's Achievement

To our knowledge, little research has focused on the reciprocal effect between parental involvement and children's reading achievement by using longitudinal designs [22]. Longitudinal studies with adolescents have found that students achieved better GPA scores when parents were more involved in school-based activities in the following years. The effect persisted over time [25,26]. Some studies have found that achievement has a stronger effect on parental involvement as vice versa [2,22].

Looking at longitudinal studies at the beginning of elementary school, results regarding the impact of school-based parental involvement seem to be inconsistent. El Nokali, Bachman, and Votruba-Drzal [27] investigated effects of parental involvement in students from first to fifth grade with one additional measurement point in third grade. The authors used standardized test scores for achievement in reading. Here, change in parents' reported parental involvement did not result in a change in reading achievement. When teachers reported parental involvement, increases in parental involvement led to children's lower reading achievement. Using a United States sample with only low-income families, Englund, Luckner, Whaley, and Egeland [28] found that a child's achievement in first grade positively predicted the change in parental involvement at third grade. Furthermore, the change in parental involvement in third grade was significantly positively correlated with the change in the child's achievement in third grade. Parental involvement and child achievement both were measured by teacher ratings [28]. Another study with an Australian sample indicated positive links between school-based involvement at grade 1 and reading achievement at grade 3 [29]. One study that focused on parental home-based involvement investigated the reciprocal relationship between the quality of home-based involvement and different child-related outcomes in the reading domain [30]. They found reciprocal relationships between children's low academic functioning and parents' use of control strategies. Low academic functioning in grade 5 led to more parental control in grade 7. Furthermore, more parental control in grade 5 led to lower academic functioning in grade 7. Similar results were found for high academic functioning and parental responsiveness.

### 1.4. Relationship between School- and Home-Based Involvement

Another important question is related to the relationship between home- and school-based parental involvement. Some studies reported a positive relationship between these two forms [31,32]. Parents who are more involved in school activities are also more effective at home-based involvement. Other studies have not found a statistically significant association between parents' home- and school-based involvement [5,33].

### 1.5. Structural Determinants of Parental School- and Home-Based Involvement

Structural determinants help to understand school- and home-based involvement, children's academic achievement and its associations [7,34]. We know that children's reading competence in Germany is determined by families' socio-economic background [35].

Children of parents who only speak German at home and who have an educational degree which allowed them to attend universities, have shown better vocabulary knowledge. Vocabulary knowledge is an important prerequisite for reading competence. Parent's socio-economic status also positively affects children's reading competence [35].

Previous studies have also found that migrated parents are less school-based involved due to lack of familiarity with the school system of the immigrated country and often have different expectations about their role in their child's education [14,36]. In more detail, some studies have underlined that migrated parents more often use formal forms of communication, whereas non-migrated parents are more involved in informal forms of cooperation with teachers [37]. Furthermore, migrated parents have been shown to support their children more often at home than non-migrated parents [5]. Scholars have explained these results with a higher academic aspiration for migrated parents compared to non-migrated parents e.g., [38].

Regarding parents' educational background, the results of previous studies underline that parents use different types of involvement based on their educational background. Manz, Fantuzzo, and Power [39] showed that parents with a higher level of educational background reported being more involved at home [28] and in school [14]. This paper's author previously found that parents with higher levels of educational background used fewer formal opportunities to make contact with teachers in their school-based involvement, such as attending parent–teacher conferences. Parents with higher levels of educational backgrounds also used informal forms to contact teachers in their school-based involvement than parents with lower levels of educational backgrounds [37].

Families' cultural capital is "understood as one's capabilities to understand and appreciate cultural manifestations" [40] (p. 174), [41] and is a facet of the stimulation dimension of the home learning environment, which includes opportunities and resources for children's exploration and learning at home [42]. Cultural capital is often operationalized as the number of books at home [43]. We know that parents with a higher number of books at home are more school- and home-based involved than parents with a lower number of books at home. As mentioned above, cultural capital, educational background, and migration background have been revealed to be important predictors of children's reading achievement [35,44–46]. Because school- and home-based involvement are effective predictors of children's achievement, it is assumed that they can mediate the relationship between structural context variables and children's outcomes [11,35,44].

## 2. The Present Study

With the goal to understand the interrelations between parental school- and home-based involvement and children's academic achievement, longitudinal studies are necessary. Such studies can provide a deeper understanding of influences over a period of time and might encourage schools, teachers, and parents to optimize their family–school partnership [19]. While prior studies often have investigated students' GPA as outcome, it is also of interest if parental involvement has an impact on the child's competence, e.g., students' development in reading. The following study aimed to investigate cross-lagged effects between parental school-based involvement, home-based involvement, and children's standardized reading achievement during the first school year of elementary school. We focused on parents' reported involvement in schools, such as volunteering and communicating as a form of school-based involvement. Regarding home-based involvement, we were interested in parents' support at home during children's learning activities.

At first, we aimed to understand the longitudinal relationship between school-based involvement and reading achievement. The first research question (RQ) is:

(RQ1) To what extent are school-based involvement and reading achievement related?

We expect that school-based involvement at the beginning of grade 1 is associated with the relative change in children's reading achievement at the end of grade 1 (H1.1). Further, we expect that reading achievement at the beginning of grade 1 is associated with school-based involvement at the end of grade 1 (H1.2). Former research has shown both

negative and positive links between school-based involvement and children's achievement. Due to this, we have no directed hypotheses.

The second research question is focused on home-based involvement:

(RQ2) To what extent are home-based involvement and reading achievement related?

We expect that home-based involvement at the beginning of grade 1 is associated with the relative change on reading achievement at the end of grade 1 (H2.1). Further, we expect that reading achievement at the beginning of grade 1 is associated with home-based involvement at the end of grade 1 (H2.2). We have not formulated a direct hypotheses because researchers have not shown a clear pattern of either positive or negative effects of home-based involvement on reading achievement and vice versa.

As previous research indicates that school- and home-based involvement are often interrelated [17,18,21], the third research question asks the following:

(RQ3) How are school-based and home-based involvement interrelated?

Based on results that have shown that parents learn about different and effective ways to support their children at home via communication with teachers, we assume that parents who are more often involved in the school at the beginning of grade 1 support their children at home more often at the end of grade 1 (H3.1).

To acquire a deeper understanding of the interrelation between school- and home-based involvement and reading achievement, it is important to consider the influence of structural context predictors on parental involvement and children's reading achievement [28]. Three main structural context predictors were investigated in the recent study: parents' migration and educational background, and families' cultural capital. The fourth research question is:

(RQ4) How are structural context predictors related to school- and home-based involvement and reading achievement.

We assume that parents with a migration background report a lower school-/home-based involvement than parents without a migration background (H4.1). Furthermore, parents with a lower level of educational background should be less involved in school and should support their children less at home (H4.2). Additionally, we expect parents with high cultural capital to report less school-based but higher home-based involvement than parents with low cultural capital (H4.3). We assume at least that children with parents who have higher levels of educational background, high cultural capital, and no migration background show better scores in reading achievement (H4.4).

Because school- and home-based involvement are more changeable predictors of reading achievement than the structural context factors, they can have a mediating role. The mediation effect will be addressed in the fifth research question:

(RQ5) Do school- and home-based involvement mediate the relationship between structural context factors (education, cultural capital, and migration) and reading achievement?

## 3. Materials and Methods

### 3.1. Procedure and Participants

A subsample of the German *LIFE-Experience Reading in Family-Study* was used to answer our research questions. The study took place in four different elementary schools which were situated in the northwestern part of Germany [45]. Data were collected from 2013 to 2018. Schools were chosen at random and contacted via the school principals. After we received consent for the study, every parent was informed. When we received the parents' permission to participate in the *LIFE-Study*, children were interviewed by a paper–pencil questionnaire (total: 60 min; 30 min reading achievement test; 30 min home literacy environment questionnaire). Parents received questionnaires through letters sent home.

Data of parents and children were collected at two measurement points during grade 1 in elementary school. The first assessment (*t1*) was confirmed some months after children were enrolled in elementary school (September until October). The second measurement point (*t2*) was at the end of grade 1 (July until August of the following year).

In total, we used a subsample of *N* = 254 parent–child dyads. Children were *M* = 6.11 (*SD* = 0.32) years old at the first measurement point. In this subset, at least 43.6% of the children were female. Further, socio-demographic information about the families is presented in Table 1.

**Table 1.** Families' Socio-demographic Characteristics.

| Families' Characteristics | *N* = 254 | |
| --- | --- | --- |
| | **Mother** | **Father** |
| Age (*M*, *SD*) | 37.88 (5.01) | 39.36 (5.45) |
| Married with father/mother (or live in a relationship) | 91.00% | 96.60% |
| Immigrant background (not born in Germany) | 26.40% | 32.40% |
| *Parents' educational background* | | |
| Low educational level without any apprenticeship | 2.4% | 2.1% |
| Low educational level | 6.1% | 16.7% |
| Middle educational level | 22.4% | 19.2% |
| High educational level | 34.7% | 26.4% |
| University degree | 34.3% | 35.6% |
| *Parents' employment* | | |
| Full-time job | 11.4% | 87.0% |
| Part-time job | 59.6% | 4.5% |
| Job-seeking | 3.7% | 1.6% |
| Homemaker | 16.3% | 1.2% |
| Other | 9.0% | 5.7% |

*3.2. Instruments*

3.2.1. School-Based Involvement

*School-based involvement* was assessed with four items from the German Parents' Involvement in their Child's School Scale [47]. Parents estimated how much they were involved in their children's school (e.g., "help the teacher during class"). The response scale ranged from 1 = *does not apply at all* to 5 = *applies absolutely*. Reliability ranged from acceptable to good at all measurement points ($\alpha(t1)$ = 0.81; $\alpha(t2)$ = 0.78).

3.2.2. Home-Based Involvement

*Home-Based involvement* was assessed with three items. Parents reported how often they supported their children at home carrying out their general and reading homework (e.g., "How often do you support your child with his reading homework."). We have developed this scale ourselves in the LIFE-Project [45]. The response scale ranged from 1 = *never or almost never* to 5 = *every day*. Reliability ranged from acceptable to good ($\alpha(t1)$ = 0.72; $\alpha(t2)$ = 0.96).

3.2.3. Structural Context Variables

We measured *structural context variables* with three indicators.

*Parents' migration background* was assessed using a dichotomous variable (0 = *at least one parent was not born in Germany* and 1 = *both parents were born in Germany*, [48]).

*Parents' educational background* was assessed asking about parents' highest educational degree [6]. Parents had to choose their highest degree of a list of five different German school tracks (1 = *low educational level without any apprenticeship*, 2 = *low educational level*, 3 = *middle educational level*, 4 = *high educational level*, and 5 = *university degree*. We calculated a dummy variable for further analyses with 0 = *low/middle educational level* and 1 = *high educational level/university degree*).

*Families' cultural capital* was operationalized using parent reports of the number of books at home with a rating scale from 1 = *0–10 books*, to 5 = *over 201 books* [48,49]. For further analysis, we also calculated a dummy variable with 0 = *less than 200 books* and 1 = *201 books and more*.

### 3.2.4. Children's Reading Achievement

A standardized group test measured *children's reading achievement (*t1*)* to indicate children's *phonological awareness* as an important reading prerequisite. Therefore, we used the German group test for early diagnosis of dyslexia (PB-LRS, [50]). The test consists of five subtasks and is based on the definition of phonological awareness by Skowronek and Marx [50]. Children need to detect rhymes, syllables, the initial sound, and the end sound of words. Furthermore, they were asked to combine different sounds to a word. For further analyses, the sum score was calculated (max. score = 50, score per subtask = 10, $\alpha$ = 0.85).

*Children's reading achievement (*t2*)* was measured by a standardized group test to indicate children's reading comprehension on two different subtasks: word comprehension and sentence comprehension (ELFE 1-6 by [51]). For further analyses, a sum score was used (max score = 100). Reliability of the standardized reading comprehension test was good ($\alpha$ = 0.85).

Means and standard deviations of all scales are presented in Table 2. Results of the confirmatory factor analyses for all items and scales are presented in Appendix A.

**Table 2.** Descriptives and Bivariate Correlations.

|  | *M (SD)* | 1 | 2 | 3 | 4 | 5 | 6 | 7 | 8 | 9 |
|---|---|---|---|---|---|---|---|---|---|---|
| (1) School-based *t1* | 1.59 (0.86) | - | | | | | | | | |
| (2) School-based *t2* | 1.77 (0.87) | 0.37 *** | - | | | | | | | |
| (3) Home-based *t1* | 4.45 (0.83) | 0.14 * | 0.18 *** | - | | | | | | |
| (4) Home-based *t2* | 4.43 (0.75) | 0.03 | 0.13 | 0.28 ** | - | | | | | |
| (5) Phonological awareness *t1* | 39.73 (6.80) | 0.10 | −0.21 ** | −0.02 | −0.23 *** | - | | | | |
| (6) Reading achievement *t2* | 23.11 (12.86) | 0.02 | −0.20 *** | −0.19 * | −0.42 *** | 0.42 *** | - | | | |
| (7) Educational background | - | −0.08 | −0.07 | 0.00 | −0.12 ** | 0.16 * | 0.28 *** | - | | |
| (8) Cultural capital | - | −0.14 * | −0.11 | −0.18 ** | −0.07 | 0.06 | 0.10 | 0.36 *** | - | |
| (9) Migration background | - | −0.10 | −0.05 | −0.06 | −0.05 | 0.06 | −0.02 | 0.16 * | 0.27 *** | - |

*Note.* * $p < 0.05$. ** $p < 0.01$. *** $p < 0.001$. *t1* = Beginning of the first school year. *t2* = End of the first school year.

### 3.3. Statistical Analyses

Statistical analyses were performed using R software (version 3.5.2, [52]) with the package psych [53] and lavaan [54].

#### 3.3.1. Model Specification

We tested our hypotheses with longitudinal cross-lagged panel modeling using half longitudinal mediation [55,56]. Due to sample size, we used manifest variables of the constructs for path modeling. With this, we specified the reciprocal relationship between parental involvement (school- and home-based involvement) and children's reading achievement. We used the following goodness-of-fit indices and cut-off criteria to evaluate whether the assumed model fit the data [56,57]: chi-square/df ($\leq$2.0 excellent model fit), comparative-fit index (CFI, 0.90–0.95 acceptable and 0.96–0.99 very good model fit), root-mean-square-error of approximation (RMSEA, 0.08–0.05 acceptable and 0.04–0.02 very good model fit), and standardized-root-mean residual (SRMR, 0.08–0.05 acceptable and 0.04–0.02 very good model fit). All analyses were conducted using maximum likelihood with robust standard errors and chi-square values (MLR).

In order to examine possible mediations between school- and home-based involvement, indirect effects were calculated, and the asymptotic confidence intervals were given (to check the significance of specific indirect effects, 95% confidence intervals based on 1000 bootstrap samples were used; [58,59]). According to Zhao, Lynch and Chen [60], the following types of indirect effects are distinguished: (1) complementary mediation—both the mediated effect and the direct effect exist and have the same sign, (2) competitive mediation—both the mediated effect and the direct effect exist and have contrary signs, (3) indirect-only mediation—a mediated effect but no direct effect exists, (4) direct-only mediation—a direct but no indirect effect exists.

All tested hypotheses are represented in Figure 1.

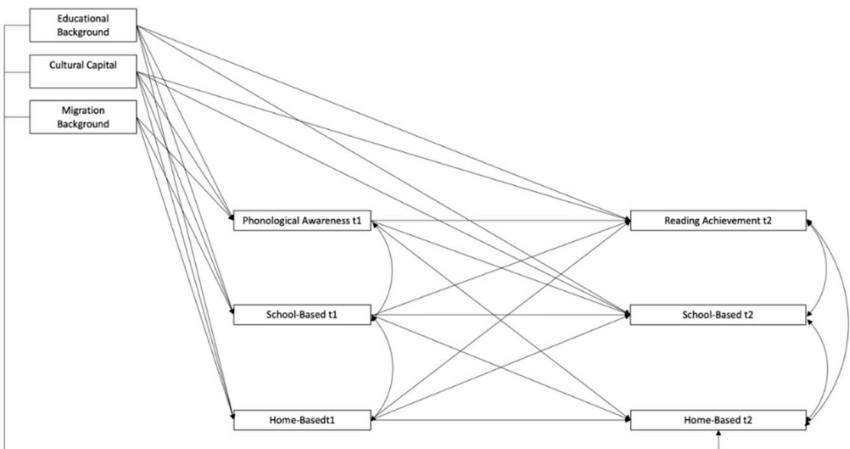

**Figure 1.** Hypothesized tested cross-lagged panel model. (*Note.* Hypothesized model with all relationships between parental involvement and children's reading achievement, and structural context variables.).

### 3.3.2. Missing Data

As mentioned above, we used a subsample of $N = 254$ parent–child dyads from the *LIFE-Study*. In total, we asked $N = 674$ children at the first measurement point and $N = 739$ children at the second measurement point. Not all children's parents participated in the study at each measurement point. In all, $N = 523$ parents participated at the first measurement point, $N = 383$ at the second. Due to the high number of missing parents at both measurement points, independent *t*-tests were performed to investigate if there were differences in children's reading achievement when a parent participated in the study (s1) or did not participate (s0). Parents of children who had a higher score in the reading achievement test at *t2* were more likely to participate in the study at *t2* ($M_{s1} = 22.63$ ($SD_{s1} = 12.29$), $M_{s0} = 20.46$ ($SD_{s0} = 12.52$), $t(727) = -2.36$, $p = 0.02$. Therefore, only full data sets of parent–child dyads over the two measurement points ($N = 254$) were used. According to Little's MCAR test, missing data in the relevant variables (min. 0%, max. 15%) in this subsample were completely at random ($\chi^2(101) = 74.00$, $p = 0.98$). Missing data were addressed using full information maximum likelihood (FIML) estimation. The nested data structure (parent–child dyads in school classes) were considered using cluster-robust standard errors [61,62].

## 4. Results

### 4.1. Model-Fit

The performed cross-lagged panel model with half longitudinal mediation (Figure 2) showed a good model-fit ($\chi^2(1) = 0.044$, $p = 0.83$; CFI = 1.000; TLI = 1.143; RMSEA = 0.000; SRMR = 0.002).

### 4.2. Reciprocal Relations between School-Based Involvement and Reading Achievement

The first research question addressed the reciprocal relationship between school-based involvement and reading achievement. Results show (see Figure 2) that phonological awareness at *t1* was significantly negatively associated to the relative change in school-based involvement at *t2*. The higher the children's phonological awareness was at the beginning of grade 1, the more parents decreased in the ranking order of school-based involvement at the end of grade 1. In summary, H1.2 can be maintained, whereas H1.1 needs to be rejected.

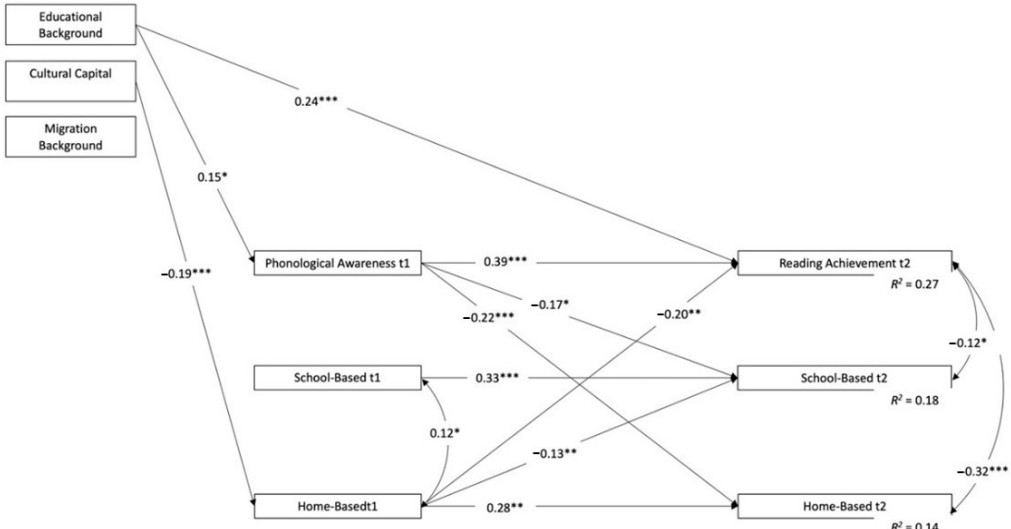

**Figure 2.** Final model for reciprocal relationship between parental involvement and children's reading achievement. (*Note.* Only significant paths were presented; * $p < 0.05$. ** $p < 0.01$. *** $p < 0.001$; Educational background: low/middle = 0; high = 1; Cultural capital: under 200 books = 0; over 200 books = 1; migration background: not born in Germany = 0; born in Germany = 1).

### 4.3. Reciprocal Relations between Home-Based Involvement and Reading Achievement

The second research question focused on the relationship between home-based involvement and reading achievement. Figure 2 shows that phonological awareness at *t1* was significantly negatively related to the relative change in home-based involvement at *t2*. Further, home-based involvement at *t1* also was significantly negatively related to the relative change in reading achievement at *t2*. First, this means that a higher score in children's phonological awareness at the beginning of grade 1 led to a decrease in the ranking order of parental home-based involvement at the end of grade 1. Second, the more often parents reported to help their children at home with school-related tasks at the beginning of grade 1, the more children decreased in the ranking order of reading achievement at the end of grade 1. By maintaining H2.1 and H2.2, we conclude a reciprocal relationship between home-based involvement and reading achievement.

### 4.4. The Relationship between School- and Home-Based Involvement

Research question 3 focused on the relationship between school- and home-based involvement. As visualized in Figure 2, home-based involvement at *t1* was significantly negatively associated with the relative change in school-based involvement at *t2*. Rejecting hypotheses H3.1, we can summarize that parents who reported a high quantity of home-based involvement at the beginning of grade 1 decreased in the ranking order of school-based involvement at the end of grade 1.

### 4.5. Mediation Analyses

Research questions 4 and 5 investigated the influence of structural context variables (migration background, educational background, cultural capital) on the relative change in reading achievement and school- and home-based involvement. Further, we examined if school- and home-based involvement mediate the relationship between structural context variables and reading achievement. Rejecting H4.1, parents' migration background was not related to school- or home-based involvement, or children's reading achievement. Partly accepting hypotheses H4.2, parents' educational background was positively related to children's phonological awareness (*t1*) and relative change in reading achievement (*t2*) but not to school- and home-based involvement. Rejecting H4.3, cultural capital was negatively related to home-based involvement at *t1*. Parents who reported possessing less than 200 books at home supported their children more often at home at the beginning of grade 1.

Regarding research question 5, no indirect effects of school- and home-based involvement in the relationship between structural context variables and reading achievement were found (Table 3).

**Table 3.** Indirect and Total Effects.

| Mediation | ß | (SE) | *p* | 95% CI |
| --- | --- | --- | --- | --- |
| School-based involvement | | | | |
| Indirect Effect | | | | |
| Educational background—school-based—Δ reading achievement | −0.00 | 0.09 | 0.86 | [−0.20; 0.16] |
| Total Effect | | | | |
| Educational background—Δ reading achievement | 0.24 | 1.40 | 0.00 | [3.64; 9.13] |
| Indirect Effect | | | | |
| Cultural capital—school-based—Δ reading achievement | −0.00 | 0.11 | 0.71 | [−0.26; 0.18] |
| Total Effect | | | | |
| Cultural capital—Δ reading achievement | 0.24 | 1.40 | 0.00 | [3.63; 9.10] |
| Indirect Effect | | | | |
| Migration background—school-based—Δ reading achievement | 0.00 | 0.10 | 0.89 | [−0.18; 0.21] |
| Total Effect | | | | |
| Migration background—Δ reading achievement | 0.24 | 1.43 | 0.00 | [3.62; 9.21] |
| Home-based involvement | | | | |
| Indirect Effect | | | | |
| Educational background—home-based—Δ reading achievement | 0.02 | 0.30 | 0.10 | [−0.99; 1.08] |
| Total Effect | | | | |
| Educational background—Δ reading achievement | 0.26 | 1.53 | 0.00 | [3.90; 9.88] |
| Indirect Effect | | | | |
| Cultural capital—home-based—Δ reading achievement | −0.00 | 0.27 | 0.68 | [−0.65; 0.42] |
| Total Effect | | | | |
| Cultural capital—Δ reading achievement | 0.24 | 1.37 | 0.00 | [3.14; 8.97] |
| Indirect Effect | | | | |
| Migration background—home-based—Δ reading achievement | 0.00 | 0.29 | 0.80 | [−0.49; 0.64] |
| Total Effect | | | | |
| Migration background—Δ reading achievement | 0.24 | 1.40 | 0.00 | [3.73; 9.23] |

## 5. Discussion

The main goal of this study was to understand the complex relationship between parental school- and home-based involvement with children's reading achievement at the beginning of elementary school. This goal addressed the research gap of the direction in the relationship between these variables over a period of time. To better understand the relationship, we investigated the impact of structural context variables (migration background, educational background, and cultural capital) on school- and home-based involvement and children's reading achievement by also testing mediation effects.

### 5.1. The Reciprocal Relationship between School- and Home-Based Involvement with Reading Achievement

First, we found a negative association between children's phonological awareness and the relative change in parental school-based involvement (RQ1). Parents with children who reached a higher test score in the standardized phonological awareness test at the beginning of grade 1 decreased in the ranking order of school-based involvement at the end of grade 1. A possible explanation for this result might be that there is no need for these kinds of parents to intensify the communication with teachers by participating more often in school events. This finding underlines the importance of children's achievement at the beginning of school and illustrates how parents may change their behavior according to their children's academic development [12,25,63].

Contrary to the result of RQ 1, we found a negative reciprocal relationship between parental home-based involvement at the beginning of grade 1 and children's reading

achievement at the end of grade 1. Similarly to school-based involvement, parents of children with higher phonological awareness test scores at the beginning of grade 1 also decreased in the ranking order of home-based involvement at the end of grade 1. If parents recognized that their children were performing well at school during the first school year and experienced their children not having reading difficulties, this might lead to adjusting their amount of home support. The support that parents with higher-achieving children offered at the beginning of grade 1 might not be needed anymore at the end of grade 1. A possible reason could be that children were able to do their (reading) homework on their own [8]. Furthermore, parents with lower-achieving children may become more concerned about their children's reading performance during the first school year in elementary school and therefore become more involved. In that case, involvement might be a response to lower performance. Future research should integrate parental concerns about children's academic development. Such investigations may explain increases or decreases in parental involvement in more detail.

Furthermore, children supported by their parents at the beginning of grade 1 more often decreased in the ranking order of reading achievement test scores at the end of grade 1. The quality of parental involvement could explain this negative effect association. It may be that in these kinds of families, parents become stressed about their child's school performance, resulting in more controlling parental behavior. Dumont et al. [30] showed that for home-based involvement control strategies of parents were linked to negative forms of learning and acquisition. It is possible that parents who show a high level of home-based involvement also tend to show a higher amount of these control strategies. This further negatively effects motivational beliefs, such as self-concept or self-efficacy, which, in turn, negatively influences children's achievement [2,3,64,65].

Regarding RQ 3, we can summarize that parents who supported their children more often at home in reading activities at the beginning of grade 1 decreased in the ranking order of school-based involvement at the end of grade 1. Similar to previous findings by the Rubach and Bonanati [5], we did not find the assumed positive association between school-based and home-based involvement [32]. The indicated negative association might illustrate that parents prioritize their type of involvement during the first year of elementary school. Further research, which needs to include more predictors of parental involvement, is needed to explain this effect. For example, parents might only have the resources of time and energy to become involved in one type of parental involvement [8].

## 5.2. Structural Context Variables as Predictors of Parental Involvement and Reading Achievement

The recent study underlines the importance of parents' educational background for children's reading achievement [35,44]. Children of parents who reported a higher level of educational background (e.g., university degree) showed better test results in phonological awareness at the beginning and reading achievement at the end of grade 1. Contrary to our hypothesis concerning the positive relationship between the number of books at home and parental home-based involvement, parents who reported to possess a lower number of books at home supported their children more often with general and reading homework. As the results of our study reflect, it is often assumed that children of parents with lower educational background or in this case, a lower number of books at home, need the support from their parents more often [38]. Furthermore, parents who possess a lower number of books may have higher educational aspirations for their children or express other opinions about their child's parenting [66]. Because we only focused on parental behaviors for involvement and not on parental expectations, further investigation is needed to understand the relationship between book amount and home-based involvement. Clarifying this effect might help to understand particular groups of parents in a better way.

## 5.3. Limitations

Next to our important findings, the current study has some limitations that need to be discussed, limiting the generalization of the results. Analyses of missing data revealed

a higher drop out of parents from the surveys when children showed lower reading test scores. The high and often not at random drop out of study participants is an often-mentioned problem of longitudinal studies [26,67] (This leads to a potential bias in the sample, which reduces the interpretability of the results. Furthermore, parental participation in a longitudinal survey about school- and home-based involvement might be one expression of parental involvement itself. It is possible that only parents who were interested in supporting their child participated in our study. Means of school-based involvement were lower ($M_{t1} = 1.59$, $SD_{t1} = 0.86$; $M_{t2} = 1.77$, $SD_{t2} = 0.83$) compared to means of home-based involvement ($M_{t1} = 4.45$, $SD_{t1} = 0.83$; $M_{t2} = 4.43$, $SD_{t2} = 0.75$). But this does not reflect parental motivation to become involved. Motivational beliefs as an important predictor of parental involvement should be included in further analyses and results of the current study need to be interpreted carefully due to this bias [21]. Other possibilities to survey the development of school- and home-based involvement could reduce the drop out. For example, parents could respond to a small battery of questions sent to them via smartphone links during the first years of school instead of answering a long paper–pencil questionnaire. For future research on family–school partnership, it seems to be important to establish more flexible methods of investigation.

In the current study, we only asked parents to report their perception of school- and home-based involvement. Results are based on self-reported data of parents. Especially for home-based involvement, social desirability may be a problem. Former research has shown differences regarding the effect of school-based involvement on achievement by using different perspectives [27]. Therefore, future studies might need to consider parents', teachers', and children's perceptions of parental involvement. In addition, more objective measures, such as the observation of parental involvement by a third independent person, may reduce potential social desirability.

Furthermore, the focus of the current study was on the longitudinal effects of parental involvement on children's achievement. Therefore, we decided only to investigate parental behaviors by using the common differentiation between school- and home-based involvement [18,20]. It is important to emphasize that parental involvement is a multidimensional construct which—next to parental behaviors—also consists of parental expectations and other forms of communication [14]. Different types of involvement may influence different aspects of children's academic achievement in a different way [10].

### 5.4. Implications and Conclusions

The current study results have drawn attention to the effects between school- and home-based involvement and children's reading achievement. The results of the study underline the hypothesis that parents adjust the type of support depending on how well their child is performing in school. Therefore, the first grade seems to be an important time to strengthen parental involvement in children's education, and establish family–school partnerships. During the first grade, teachers and parents can establish constructive structures concerning their communication. Our results show that children's achievement is an important aspect which determines the establishment of parental involvement. One idea for teachers might be to talk about children's school success with parents in order to be transparent and communicate suitable parental involvement strategies.

**Author Contributions:** S.B., Conceptualization, methodology, formal analysis, investigation, writing—original draft, writing—review & editing. C.R., Conceptualization, writing—review & editing. All authors have read and agreed to the published version of the manuscript.

**Funding:** This research received no external funding.

**Institutional Review Board Statement:** The study was conducted according to the guidelines of the Declaration of Helsinki, and approved by the Institutional Review Board (or Ethics Committee) of Paderborn University.

**Informed Consent Statement:** Informed consent was obtained from all subjects involved in the study.

**Data Availability Statement:** The data presented in this study are available on request from the corresponding author. The data are not publicly available due to the form anonymity which was promised in the form of consent the subjects obtained.

**Acknowledgments:** We like to thank all the schools and families who participated in the study.

**Conflicts of Interest:** The authors declare no conflict of interest.

## Appendix A

**Table A1.** Factor Loadings of All Scales and Used Items.

| Scales and Items | λ (Model *t1*) | λ (Model *t2*) |
|---|---|---|
| **School-Based Involvement** | | |
| Talk with the teacher if he or she wishes it (next to formal parent-teacher conferences). | 0.66 | 0.43 |
| Volunteer in school library. | 0.77 | 0.67 |
| Help the teacher during class. | 0.81 | 0.90 |
| Participate in class. | 0.71 | 0.79 |
| **Home-Based Involvement** | | |
| How often do you help your child with his homework? | 0.32 | 0.51 |
| How often do you helpf your child with his reading homework? | 1.22 | 0.99 |
| How often do you help your child with reading? | 0.57 | 0.75 |
| **Children's Reading Achievement (*t1*)** | | |
| Rhymes detection | 0.60 | - |
| Syllables detection | 0.37 | - |
| Initial sound detection | 0.71 | - |
| Sound combination | 0.42 | - |
| End sound detection | 0.71 | - |
| **Children's Reading Achievement (*t2*)** | | |
| Word comprehension subtask | - | 0.92 |
| Sentence comprehension subtask | - | 0.86 |

*Notes.* All factor loadings were significant and $p < 0.000$; $t1$ = Beginning of the first school year; $t2$ = End of the first school year; $t3$ = Middle of the second school year; Model-Fit for Model $t1$: CFA = 0.949, TLI = 0.934, RMSEA = 0.053, SRMR = 0.065; Model-Fit for Model $t2$: CFA = 0.984; TLI = 0.977, RMSEA = 0.047, SRMR = 0.046.

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
