# Peer review of "Reciprocal Relationship between Parents’ School- and Home-Based Involvement and Children’s Reading Achievement during the First Year of Elementary School"

_societies, doi:10.3390/soc12020063_

Round 1

Reviewer 1 Report

The authors report an interesting study on the effects of different types of parental involvement on children's reading achievement.  I recommend some minor revisions (see below)

General comments:

  • My main criticism is that I do not think you can claim that children's achievement predicts parental involvement. It is certainly associated, but there may be other explanations for why children with low reading achievement had higher parental involvement. Perhaps parents are concerned and become more involved as a response to lower performance. However, as you mention, it could also be that parental involvement causes poorer reading achievement, possibly by stifling the child's independent learning or creating a stressful, controlling learning environment. I also wonder whether the quality of parental involvement matters since the questionnaires mostly measured quantity.
  • Copy editing would improve clarity and readability.

Specific Comments

  • Page 3: The sentence "Low academic functioning in grade 5 led to more parental control in grade 7 and vice versa" is confusing and implies that more parental control in grade 7 led to low academic functioning in grade 5, which is not possible. Perhaps you meant that the relationship between parental control and academic functioning was reciprocal, and more parental control in grade 5 also led to low academic functioning in grade 7.
  • Page 4: Please specify the direction of the relationship between parents' migration status and reading competence/vocabulary. Based on your explanation, I inferred that children with parents from another country had higher reading and vocabulary scores, perhaps because they have higher educational aspirations for their kids. However, on Page 5, you assume that children whose parents have no migration background have better reading scores (H4.4).
  • Page 4: Could you define the concept of cultural capital? I understand how it was measured, but I am not sure if this is meant to serve as a proxy for parental education, socioeconomic status, or some kind of cultural competence.
  • Page 6: You mention that children were interviewed with a 60-minute questionnaire. Was this just the testing session, or what did the questionnaire include?
  • Page 7: It was difficult to read Table 2 split across pages
  • Page 13: You might add the limitations of self-report data or provide references that they are in line with objective measures so that parents do not inflate measures of their involvement.

Author Response

Dear guest editor Dr. Laura Nathans,

Dear Editor-in-Chief Dr. Gregor Wolbring,

Dear April Teng,

Dear reviewers,

Thank you and the reviewers very much for the helpful feedback and valuable recommendations.

In our revision, we replied to the reviewers’ questions and comments and describe how we revised the paper based on each comment. Citations from the revised manuscript are also provided.

We highly appreciate the opportunity to revise the manuscript and hope that you will consider our paper for publication in societies.

Sincerely,

the authors

Reviewer #1

Comment:

My main criticism is that I do not think you can claim that children's achievement predicts parental involvement. It is certainly associated, but there may be other explanations for why children with low reading achievement had higher parental involvement. Perhaps parents are concerned and become more involved as a response to lower performance. However, as you mention, it could also be that parental involvement causes poorer reading achievement, possibly by stifling the child's independent learning or creating a stressful, controlling learning environment. I also wonder whether the quality of parental involvement matters since the questionnaires mostly measured quantity.

Response:

Thank you very much for this comment. Our research is guided by Hoover-Dempsey and Sandler’s (1995, 1997) model of parental involvement process and Eccles and Harold’s (1996) model of influences on and consequences of parent involvement in schools that post the interrelation between parental involvement and children’s performance in school. Therefore, research can assume both that performance impacts involvement and vice versa. We, however, agree with you that there are further explanatory factors for the negative association between children’s reading performance and parental involvement. We t added the aspect about the parental concerns regarding their children’s reading performance in the discussion section. We also revised the whole manuscript carefully according to sentences where we used the term predict.

We further agree that the quality of parental involvement matters for children’s positive development in school. We therefore, added in the discussion that the quality of involvement might be another factor that future research needs to take into account

Page. 12: “Furthermore, parents with lower-achieving children may become more concerned about their children’s reading performance during the first school year in elementary school and therefore, become more involved. In that case, involvement might be a response to lower performance. Future research should integrate parental concerns about children’s academic development. Such investigations may explain increases or decreases of parental involvement in more detail.”

Page 12: “Furthermore, children supported by their parents at the beginning of grade 1 more often decreased in the ranking order of reading achievement test scores at the end of grade 1. The quality of parental involvement could explain this negative effect association. Maybe, in these kinds of families, parents get stressed about the child’s school performance, resulting in more controlling parental behavior. Future research needs to take into account both, the quality and the quantity of parental involvement.”

Comment:

Copy editing would improve clarity and readability.

Response:

The manuscript was carefully edited.

Specific Comments

Comment:

Page 3: The sentence "Low academic functioning in grade 5 led to more parental control in grade 7 and vice versa" is confusing and implies that more parental control in grade 7 led to low academic functioning in grade 5, which is not possible. Perhaps you meant that the relationship between parental control and academic functioning was reciprocal, and more parental control in grade 5 also led to low academic functioning in grade 7.

Response:

Thank you for the comment. We have corrected this sentence and now present the results of the study more clearly.

Page 3.: “One study that focused on parental home-based involvement investigated the reciprocal relation between the quality of home-based involvement and different child-related outcomes in the reading domain (Dumont, Trautwein, Nagy & Nagengast, 2014). They found reciprocal relations between child’s low academic functioning and parents’ use of control strategies. Low academic functioning in grade 5 led to more parental control in grade 7. Furthermore, more parental control in grade 5 led to lower academic functioning in grade 7.”

Comment:

Page 4: Please specify the direction of the relationship between parents' migration status and reading competence/vocabulary. Based on your explanation, I inferred that children with parents from another country had higher reading and vocabulary scores, perhaps because they have higher educational aspirations for their kids. However, on Page 5, you assume that children whose parents have no migration background have better reading scores (H4.4).

Response:

We cleared the previous findings concerning the relation between migration background and reading competence as well as between educational background and reading competence and hope our H4.4 is now more understandable.

Page 4: “We know that children’s reading competence in Germany is determined by families’ social-economic background (McElvany, Becker, & Lüdtke, 2009). Children of parents who only speak German at home and who have an educational degree which allow them to attend universities, showed better vocabulary knowledge. Vocabulary knowledge is an important prerequisite of reading competence. Parent’s socio-economic status also positively affected children’s reading competence (McElvany et al., 2009).”

Comment:

Page 4: Could you define the concept of cultural capital? I understand how it was measured, but I am not sure if this is meant to serve as a proxy for parental education, socioeconomic status, or some kind of cultural competence.

Response:

We have added a definition of the concept of cultural capital.

Page 4: “Families’ cultural capital is “understood as one’s capabilities to understand and appreciate cultural manifestations” (Senkbeil, 2018, p. 174; Hollingworth, Mansaray, Allen & Rose, 2011) and is a facet of the stimulation dimension of home learning environment which includes opportuinities and resources for children’s exploration and learning at home (Bradley, Pennar, Fuligni & Whiteside-Mansell, 2019). Cultural capital is often operationalized as the number of books at home (Bourdieu, 1983).”

Comment:

Page 6: You mention that children were interviewed with a 60-minute questionnaire. Was this just the testing session, or what did the questionnaire include?

Response:

Thank you for the question. We added some more details about the children’s interview. The first session to test children’s reading performance was 30 minutes long and the questionnaire using psychological instruments took another 30 minutes.

Page 6: “children were interviewed by a paper-pencil questionnaire (total: 60 minutes; 30 minutes reading achievement test; 30 minutes home literacy environment questionnaire).”

Comment:

Page 7: It was difficult to read Table 2 split across pages

Response:

We agree with the reviewers’ comment. We hope Table 2 is now presented on one page.

Comment:

Page 13: You might add the limitations of self-report data or provide references that they are in line with objective measures so that parents do not inflate measures of their involvement.

Response:

Thank you for the suggestion. We added some limitations about self-report data concerning the parents reports about their parental involvement.

Page 13: “Results are based on self-report data of parents. Especially for home-based involvement social desirability may be a problem. Former research has shown differences regarding the effect of school-based involvement on achievement by using different perspectives (El Nokali et al., 2010). Therefore, future studies might need to consider parents’, teachers’ and children’s perceptions of parental involvement. Also, some more objective measures, such as the observation of parental involvement by a third independent person may reduce potential social desirability.”

Reviewer 2 Report

The article deals with an important topic of educational sciences and results are clearly presented. It would be interesting to analyse other forms of shared responsibility between family and school, bypassing the bias due to the use of self report instruments.

Author Response

Dear guest editor Dr. Laura Nathans,

Dear Editor-in-Chief Dr. Gregor Wolbring,

Dear April Teng,

Dear reviewers,

Thank you and the reviewers very much for the helpful feedback and valuable recommendations.

In our revision, we replied to the reviewers’ questions and comments and describe how we revised the paper based on each comment. Citations from the revised manuscript are also provided.

We highly appreciate the opportunity to revise the manuscript and hope that you will consider our paper for publication in societies.

Sincerely,

the authors

Reviewer #2

Comment:

The article deals with an important topic of educational sciences and results are clearly presented. It would be interesting to analyse other forms of shared responsibility between family and school, bypassing the bias due to the use of self report instruments.

Response:

Thank you very much for your comment and review. You are totally right, that there is a possibility of social desirability in the parents’ self-reports about parental involvement. We extended the limitations section concerning this aspect and added some opportunities for more objective measurements.

Page 13: “Results are based on self-report data of parents. Especially for home-based involvement social desirability may be a problem. Former research has shown differences regarding the effect of school-based involvement on achievement by using different perspectives (El Nokali et al., 2010). Therefore, future studies might need to consider parents’, teachers’ and children’s perceptions of parental involvement. Also, some more objective measures, such as the observation of parental involvement by a third independent person may reduce potential social desirability.”